# Osteopontin Concentration in Prostates Fractions: A Novel Marker of Sperm Quality in Dogs

**DOI:** 10.3390/vetsci10110646

**Published:** 2023-11-08

**Authors:** Koray Tekin, Efe Kurtdede, Berrin Salmanoğlu, Ongun Uysal, Calogero Stelletta

**Affiliations:** 1Department of Reproduction and Artificial Insemination, Faculty of Veterinary Medicine, Ankara University, Ankara 06110, Turkey; tekin.koray@hotmail.com (K.T.); onuysal@veterinary.ankara.edu.tr (O.U.); 2The Department of Biochemistry, Faculty of Veterinary Medicine, Ankara University, Ankara 06110, Turkey; ekurtdede@ankara.edu.tr (E.K.); bsalman@veterinary.ankara.edu.tr (B.S.); 3Department of Animal Medicine, Production and Health, University of Padova, Legnaro, 35020 Padova, Italy

**Keywords:** elisa, malakli, osteopontin, motility, morphology

## Abstract

**Simple Summary:**

In our study, we investigate the relationship between sperm quality and osteopontin (OPN) in Malakli shepherd dogs, collecting sperm samples from 39 male dogs of various ages. We discovered significant differences in OPN levels between ejaculate fractions, with lower OPN concentrations being linked to poorer sperm motility and morphology. Our findings have the potential to significantly impact dog breeding and fertility practices, offering valuable insights to predict and enhance sperm quality for the future. This research not only advances animal reproduction science, but also benefits veterinary medicine and stud selection, ultimately contributing to the well-being of dog breeders’ society.

**Abstract:**

This study aimed to investigate the relationship between the sperm quality and the osteopontin (OPN) concentration in the prostates of Malakli shepherd dogs. Ejaculates were collected once by digital manipulation from 39 male dogs aged between 2 and 4 years and older. The first and third fractions of the ejaculate were centrifuged at 5000× *g* for 30 min, and supernatants were stored at −80 °C for further analysis of OPN using a double-antibody sandwich method (SEA899CA, Cloude-Clone Corp, Houston, TX, USA). Meanwhile, the second fractions were evaluated for sperm motility, concentration, viability, and rate of abnormal spermatozoa (head, acrosome, midpiece and tail abnormalities). The average concentration of OPN was 8.7 ± 5.2 ng/mL, and it differed significantly between the 1st 10.4 ± 5.3 ng/mL and 3rd 7.4 ± 5 ng/mL fractions. According to ROC (receiver operating characteristic curve) analysis, the OPN concentration had a better diagnostic ability for sperm motility (*p* < 0.001) than for the rate of abnormal spermatozoa (*p* < 0.05). Additionally, the OPN concentration was negatively correlated with poor sperm morphology and motility. In conclusion, the OPN concentration in prostate-derived secretions may be a possible marker of sperm quality in dogs. Further research could explore the involvement of OPN in sperm motility during cryopreservation and in vivo fertility.

## 1. Introduction

Seminal plasma (SP) proteins play several important roles in the reproductive system, including controlling the motility, vitality, and membrane stability of sperm by modifying the structure of the sperm membrane. The addition or removal of SP has been shown to have either a beneficial [1] or detrimental [2] effect in different species. OPN is a highly phosphorylated extracellular matrix glycoprotein that interacts with membrane proteins through its integrin-binding Arg-Gly-Asp sequence and plays a role in cell–cell interaction and tissue remodeling [3,4,5,6,7]. It also has various functions in male reproductive tissues and cells, such as stabilizing the spermatozoal membrane [3], promoting motility [8], facilitating capacitation [9], spermatogenesis and spermatozoa function [10,11], and contributing to inflammatory responses such as prostate cancer [12].

OPN is an SP protein that is involved in bone repair, tissue modeling, homeostatic defense mechanisms, and tumor encapsulation. OPN is found on the surface of various reproductive tissues, such as the epithelial cells of the prostate [13], seminal vesicles [14], ampulla, Sertoli cells [15], and epididymis [16] for the regulation of spermatogenesis. During gestation, OPN, also known as secreted phosphoprotein-1 (SPP1), is expressed by germ cells in the seminiferous tubules and helps prepare the testes for postnatal life [17]. OPN has also been linked to fertility in various species [18]. It has been shown to be involved in the regulation of testosterone synthesis in the testes [16] and has been suggested to play a role in the development of certain types of testicular cancer and cryptorchidism [19]. In addition, OPN is involved in the immune response of the testes and may play a role in the development of certain autoimmune conditions that affect the male reproductive system. Thus, OPN has many potential biotechnological applications, such as serving as a biomarker for fertility [8], prostate cancer [12], nephrolithiasis, and various chronic inflammatory diseases.

The identification and understanding of the roles of OPN in sperm motility, morphology, and signaling pathways could lead to the development of a measurement method for patients suffering from sperm-related infertility problems and freezing. Canine OPN identification, including key molecular biomarkers, could be an excellent model for studying complex diseases in humans [20]. In our study, we chose the Malakli shepherd dog breed due to its physiological similarity to humans, notably in body weight and prostate size. We focused on different age categories, particularly dogs under 2, to investigate a period with a minimal risk of prostate-related diseases and optimal sperm quality; beyond age 4, there is an increased risk of prostate disorders.

## 2. Material and Methods

### 2.1. Animals and Semen Collection

Thirty-nine male Malaklı shepherd dogs aged between 2 or 4 and above were used in the present study. The animals were bred in private breeder ownership and registered to the Malakli breeders association. Ejaculates were collected once from every male by digital manipulation using glass collection tubes for the cryopreservation process. Fractions were carefully collected in three parts: prostatic semen fractions (1st and 3rd fractions) and a sperm rich fraction (2nd fraction) for use in further evaluations. Animal experiments were conducted according to ethical principles, and this study was approved by the animal ethics committee of Ankara University, Turkey (2015-21-230).

### 2.2. Semen Processing and Evaluation

The volume of each fraction (mL) was determined using graded conical tubes. The first and third fractions were centrifuged in 5000× *g* for 30 min, and supernatants were stored at −80 °C for further analysis, whereas second (sperm rich) fractions were evaluated conventionally in terms of sperm motility (%) by a computer aid sperm analyzer (sperm class analyser, Microptics^®^, Barcelona, Spain), and the concentration (×10^6^) was determined by the hemocytometry method. Abnormal spermatozoa (%) were evaluated by using sperm blue (Microptics^®^) solution, and 200 spermatozoa were counted by the SCA morphology setup for dogs [21].

### 2.3. Determination of OPN Levels in Seminal Plasma

Osteopontin concentrations were determined using an ELISA kit (SEA899CA, Cloude-Clone Corp, Houston, TX, USA) by double-antibody sandwich ELISA method. The assay was performed using a competitive enzyme immunoassay technique. The sample and OPN-horseradish peroxidase (HRP) conjugate were incubated in an anti-OPN antibody pre-coated plate. A substrate for the enzyme was then added. Finally, a stop solution was added to stop the reaction. The color was measured by a spectrophotometer at 450 nm in a microplate reader. A standard curve was plotted that related to the concentration of standards. The OPN concentration in each sample was interpolated from this standard curve. Samples were measured with a standard curve.

### 2.4. Statistical Analysis

The motility and total abnormal spermatozoa rate were used to determine a cut-off point at the OPN level. The motility level was categorized as above and below 50%, and the total abnormal spermatozoa rate was categorized as below and above 30% [22]. ROC analysis was performed using the MedCalc (9.2.0.1) package program. Pearson’s correlation analysis was used to evaluate the relationship between the OPN concentration and semen parameters while considering the distribution properties of the variables. Probability values of less than 0.05 and 0.001 were considered significant unless otherwise noted and were performed by using Stata 12.1 MP4 statistical software.

## 3. Results

The collected data from 39 dogs were divided into two groups, where the OPN cut-off value (5.62 ± 0.09) showed a statistical significance between the groups. The average concentration of OPN was 8.7 ± 5.2 ng/mL and differed between the 1st and 3rd fraction, being 14.49 ± 5.3 ng/mL and 3rd 6.64 ± 5 ng/mL, respectively, independently of the age group (Table 1).

The median and range of motility were 79.8 ± 12.5 and 93.2–64.2 in the younger age group and 66.5 ± 20.10 and 55.6–4 in the older age group, respectively; for morphologically abnormal spermatozoa, the median and range were 15.02 ± 9.2 and 51.9–6.42 in the young group and 30.9 ± 17.45 and 8.6–65.4 in the older age group, respectively. The volumes of the 1st and 3rd fractions were observed to be close to each other in different age groups, as expected (young group 1st 3 ± 2; 3rd 22 ± 9.8; older group 1st 2.5 ± 1.02; 3rd 18.5 ± 9). Hence, there were no differences according to their average body weight. OPN concentrations differed statistically between the 2 and 4+ year categories (*p* < 0.046). Sperm motility, concentration and a morphologically normal spermatozoa ratio were improved in terms of quality among all categories (Table 1).

Total morphological disorders were found to be higher in dogs aged 4 years and older. However, when the morphological disorders were examined in more detail, head and acrosome-related anomaly values did not differ between age groups, while the middle part, tail and cytoplasmic droplet ratios showed values 2–3 times higher in the 4-year-old and older group than in the younger age group.

In the ROC analysis, the AUC (Area Under the Curve) value of sperm motility was 0.951 with a sensitivity of 0.962 and specificity of 92%. The morphology AUC value was 0.796 with a sensitivity of 0.72 and specificity of 90% (Table 2). 

The level of motility was found to be the strong determinant of the OPN cut-off (*p* < 0.001), and the total abnormality level was lower (*p* < 0.05). According to the ROC analysis, it has a better diagnostic ability to identify the semen quality by motility (AUC = 0.985, *p* < 000.1) than total abnormal spermatozoa rate (AUC = 0.796, *p* < 0.05) (Figure 1).

Further, when the correlation relationship between spermatological parameters with the 1st and 3rd fractions was examined, a negative correlation was found with acrosome (r = −0.493; *p* < 0.05) and mid-piece (r = −0.711; *p* < 0.01) anomalies, and a positive strong correlation was observed with motility (r = 0.763; *p* < 0.01) and first fraction OPN values (Table 3).

Statistically, no strong negative or positive correlations were found between fraction 3 and any spermatological parameters. However, as in the 1st fraction, a strong negative correlation was found between fraction 3 and mid-piece anomalies (r = −0.523; *p* > 0.05), along with a high level of positive correlation with semen concentration, and a moderate positive correlation with motility (r = −0.355; *p* > 0.05) (Table 4).

## 4. Discussion

In this study, the effects of OPN concentration on sperm parameters in canine semen fractions 1 and 3 were investigated. It was found that the OPN concentration in prostate fractions could be a potential fertility and sperm quality marker. According to ROC analysis results, average prostate OPN concentrations of individuals with a sperm motility of 50% and above had a strong diagnostic value (*p* < 0.001), while the total abnormal spermatozoa had a low diagnostic value (*p* < 0.05). However, when different morphological disorders were classified in more detail and evaluated by correlation analysis, a strong negative correlation was detected between mid-piece (*p* < 0.01) and acrosome (*p* < 0.05) anomalies and OPN concentrations.

### 4.1. Exploring OPN’s Role in Sperm Motility and Morphology

Our study revealed a link between OPN levels in the fraction of semen and sperm quality. Previous research by Lee et al. [23] demonstrated a similar connection, highlighting OPN’s protective role against apoptosis and oxidative damage in sperm cells. Abedin et al. [8]) investigated deeper, elucidating specific pathways, such as MAP kinase and phosphoinositide 3-kinase, emphasizing OPN’s influence on sperm motility. In agreement with these established findings, our study further substantiates this relationship, revealing that higher OPN levels in the first fraction are associated with reduced sperm motility. This finding is in harmony with several studies in dogs, bulls, and boars, which have consistently shown a strong connection between sperm motility, fertility, and OPN levels [8,24,25]. These collective observations underscore the pivotal role of OPN in sperm functionality and its potential significance in reproductive studies.

The reason behind this negative correlation might be OPN’s activity in the tip of sperm cells, as observed by Luedtke et al. [16]. This activity, which involves marking microvilli, endocytic vesicles, and endosomes, underscores OPN’s role in removing calcium from the epididymal lumen. By preventing calcium buildup, OPN helps maintain sperm motility. Our findings not only confirm the established connections between OPN levels, anormal spermatozoa, and motility, but also provide valuable insights into how OPN’s regulation of calcium in the epididymis directly influences sperm movement. When buffalo semen is frozen, adding SP alone harms specific sperm characteristics. However, supplementing SP with OPN improves buffalo sperm morphology by safeguarding the membrane and key parts, such as axonemes and acrosomes, and by regulating nitrite oxide synthesis [26]. Our findings revealed a strong negative correlation between OPN concentrations and mid-piece (*p* < 0.01) and acrosome (*p* < 0.05) anomalies. This discovery highlights the specific aspects of sperm morphology influenced by OPN levels. Understanding these correlations can provide valuable insights into sperm quality assessment.

### 4.2. Comparative İnsights of OPN on Fertility in Other Species

Studies in various animals, including dogs, bulls, and boars, have highlighted the connection between sperm motility, fertility, and OPN levels [8,24,25]. Furthermore, the supplementation of seminal plasma with OPN has proven beneficial in preserving sperm morphology, emphasizing the broader applicability of our findings [26] (26Viswam et al., 2020). In boars, a study examined different sperm fractions (sperm-peak, sperm-rich, and sperm-free) and their relation to in vivo fertility. The sperm-rich fraction with higher OPN-9 and 12 levels was linked to lower total born, fertility, and farrowing rates per AI dose (*p* < 0.05), while OPN-70 did not show this correlation. Interestingly, the less fertile group had a negative correlation with OPN-70 in the sperm peak fraction regarding the total born (r = −0.47, *p* = 0.02) and fertility indexes (r = −0.47, *p* = 0.03). However, OPN levels in sperm-rich and sperm-free fractions did not correlate with pregnancy rates. Differences in OPN concentration might be due to dilution rates, especially in boars where ejaculates have a high volume but low concentration [27]. Lower OPN concentrations between epididymal and ejaculated semen, as seen in Erikson et al.’s [9] study, contribute to this variance. Additionally, the composition of accessory gland fluids during ejaculation varies in different species, such as humans, horses, and pigs, which can impact OPN levels. OPN is crucial because it prevents the formation of calcium oxalate crystals, which is essential for sperm quality. Notably, boar semen has a high citric acid content, unlike dog semen, which helps maintain suitable OPN levels. Our observation that older males have lower OPN concentrations supports this idea.

OPN plays a significant role in fertility among horses [18], cattle [15], and camelids [28]. Researchers studying dogs found interactions between OPN and sperm proteins. In their study, they identified two specific OPN isoforms (77.2 kDa and 15.6 kDa) by western blot. These OPN isoforms were also found in the sperm membrane. It is believed that OPN interacts with sperm during ejaculation, forming complexes with integrins, CD44, and glycosaminoglycans until reaching the fertilization site. While OPN’s role as a fertility marker is well-established in bulls, stallions, and camelids, its relationship with canine fertility needs more research. According to Rodriguez et al. [10], OPN is associated with sperm development during spermatogenesis and also plays a role at the fertilization site. Previous studies in stallions have shown positive effects of higher levels of OPN in seminal plasma on fertility. Additionally, research conducted by Baruah et al. [29] highlighted the significant role of OPN in the cryopreservation process. In their study, OPN levels were positively correlated with various sperm characteristics. Other studies involving buffalo and bull semen combined with different OPN levels have shown improved viability and capacitation values.

Researchers have extensively studied the proteins in SP because they profoundly affect sperm quality and fertility. These proteins interact with sperm and play crucial roles throughout the sperm’s journey. One important protein, OPN, acts as a cytokine, influencing cell function through receptors. In dogs, the prostate gland secretes a significant portion of the ejaculate, and this secretion strongly influences sperm. Studies have shown that seminal plasma proteins in dogs are linked to both sperm quality and fertility [5,30]. Understanding these proteins’ functions is essential for comprehending canine reproductive physiology.

### 4.3. Potential Implications and Future Research

These findings emphasize the importance of OPN in reproductive processes, including sperm development, fertilization, and cryopreservation, across various animal species. Further research in canines could provide valuable insights into understanding and enhancing fertility in dogs. In summary, OPN concentration in the prostate fraction could potentially serve as a marker for sperm motility in dogs. Further research should explore how OPN affects sperm motility and study its impact when added to semen and extenders used in cryopreservation and in vitro production. Additionally, studies could investigate how OPN is involved in protecting sperm cells through antiapoptotic or antioxidant pathways. Understanding how OPN influences various aspects of sperm, such as motility, mitochondrial potential, membrane functionality, and DNA integrity, could guide its supplementation in semen and different extenders, potentially improving the success rate of cryopreservation and in vitro production techniques.

## Figures and Tables

**Figure 1 vetsci-10-00646-f001:**
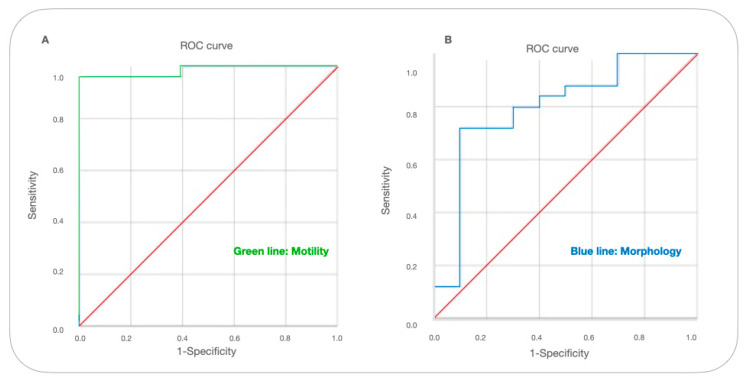
ROC curve for cut-off levels of OPN in terms of motility and abnormal spermatozoa levels. A receiver operating characteristic (ROC) curve was used to identify the levels of OPN cutoff that best predicted (**A**) motility (below 50%) and (**B**) abnormal semen (above 30%) parameters based on sensitivity and specificity.

**Table 1 vetsci-10-00646-t001:** Mean (X) and standard error (SE) values of the spermatological parameters.

Parameters	Young Group		Older Group	
Age (year)	1.5 ± 0.3 (*n* = 19)		4.7 ± 1.3 (*n* = 20)	
Fractions	1st	3rd	1st	3rd
Volume (mL)	3 ± 2	22 ± 9.8	2.5 ± 1.02	18.5 ± 9
OPN concentration (ng/mL)	17.06 ± 7.16	8.27 ± 4.54	11.58 ± 5.23	6.10 ± 4.05
OPN average (ng/mL)	1st 10.4 ± 5.3		3rd 7.4 ± 5	
Motility %	79.8 ± 12.5		66.5 ± 20.10	
Concentration × 10^6^	311.8 ± 68.3		279 ± 69.41	
Dead spermatozoa %	14.68 ± 7.7		18.5 ± 8.9	
Abnormal Spermatozoa %	15.02 ± 9.2		30.9 ± 17.45	
Head Abnormality %	2.8 ± 2.1		4.2 ± 3.4	
Acrosome defects %	2.2 ± 3.5		2.8 ± 3.2	
Mid-piece defects %	2.09 ± 2.26		7.8 ± 6.2	
Tail %	5.9 ± 4		11.5 ± 8.1	
Cytoplasmic Droplet %	2.8 ± 2.2		7.5 ± 7.9	

**Table 2 vetsci-10-00646-t002:** AUC comparison of sperm motility and morphology (motility values lower than 45% and abnormal morphology over 30%).

	AUC	SE	SP	P	95%Cl
Motility	0.985	0.962	0.95	0.001	(0.951–1)
Morphology	0.796	0.72	0.9	0.05	(0.619–0.973)

**Table 3 vetsci-10-00646-t003:** Correlation efficiency between fraction-1 OPN level and spermatological parameters.

	OPN	FR-1	FR-3	CON	Dead	Norm	ACR	Head	Mid-P	Tail	C-Drop	ABN	MOT
OPN	1	−0.170	−0.092	0.292	−0.161	0.418	−0.493 *	−0.359	−0.711 **	−0.304	−0.303	−0.417	0.763 **
Volume−1		1	−0.205	0.247	−0.225	0.054	−0.070	0.020	0.067	−0.137	0.136	−0.052	0.070
Volume−3			1	−0.139	0.400	0.159	0.049	−0.107	0.038	−0.203	−0.131	−0.157	−0.066
CON				1	−0.262	0.241	−0.141	−0.284	−0.305	−0.082	0.029	−0.233	0.178
Dead					1	−0.357	0.477 *	0.243	0.575 *	0.147	0.208	0.353	−0.403
Normal						1	−0.553 *	−0.351	−0.510 *	−0.874 **	−0.872 **	−1.000 **	0.293
ACR							1	0.583 *	0.560 *	0.240	0.268	0.546 *	−0.574 *
Head								1	0.259	−0.026	0.010	0.348	−0.306
Mid-P									1	0.343	0.364	0.508 *	−0.758 **
Tail										1	0.878 **	0.879 **	−0.172
Cyto-Drop											1	0.876 **	−0.125
Abnormal												1	−0.288
Motility													1

* Correlation is significant at the 0.05 level (2-tailed). ** Correlation is significant at the 0.01 level (2-tailed).

**Table 4 vetsci-10-00646-t004:** Correlation efficiency between fraction-3 OPN levels and spermatological parameters.

	OPN	FR-1	FR-3	CON	Dead	Normal	ACR	Head	Mid-P	Tail	Cyto-Drop	Abnormal	Motility
OPN	1	0.713	0.144	0.497	−0.282	0.173	0.031	0.133	−0.523	0.316	−0.283	−0.307	0.355
Volume−1		1	−0.064	0.743	−0.656	0.846	0.635	−0.089	−0.963 *	−0.364	−0.733	−0.896	0.898
Volume−3			1	−0.843	0.728	−0.446	0.716	−0.248	0.488	0.317	0.571	0.505	−0.769
CON				1	−0.899 *	0.770	−0.304	0.089	−0.873	−0.455	−0.827	−0.841	0.961 **
Dead					1	−0.874	0.044	−0.297	0.870	0.749	0.978 **	0.908 *	−0.865
Normal						1	0.252	−0.128	−0.929 *	−0.836	−0.919 *	−0.989 **	0.860
ACR							1	−0.089	−0.187	−0.300	−0.165	−0.200	−0.225
Head								1	0.065	−0.083	−0.264	0.070	−0.149
Mid-P									1	0.599	0.901 *	0.971 **	−0.897 *
Tail										1	0.804	0.767	−0.555
Cyto-Drop											1	0.942 *	−0.812
Abnormal												1	−0.898 *
Motility													1

* Correlation is significant at the 0.05 level (2-tailed). ** Correlation is significant at the 0.01 level (2-tailed).

## Data Availability

The datasets and models used or analyzed during the current study are available from the corresponding author upon reasonable request. The availability of the data and models is subject to any restrictions imposed by the Scientific and Technological Research Council of Turkey (TUBITAK) regarding data protection and intellectual property rights.

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
