# Peer review of "Osteopontin Concentration in Prostates Fractions: A Novel Marker of Sperm Quality in Dogs"

_vetsci, 2023, doi:10.3390/vetsci10110646_

Round 1

Reviewer 1 Report

Comments and Suggestions for Authors

Line 13: osteopontin does not need to be capitalized and there should be an s after prostate.

Line 14:  “Shepherd dog.” should be “shepherd dogs.” Correct capitalization throughout text.

Line 17: Sandwich not capitalized.

Line 18: not sure if we need to have ELISA spelled out here, editor?

Line 20: Head not capitalized.

Line 23: ROC should be defined, not common.

Line 32 spelling…

Line 39 remove the word ‘text’.

Line 44: recommend a tie in term or sentiment that OPN is an SP, for flow of thought.

Line 53: this line should be introduced earlier with the abbreviation (eg the abbreviation is always defined initially. It would also help the flow as mentioned above. Maybe consider switching both paragraphs?

Line 50 change to spermatozoal

Line 60 spell out SPP1 and show relevance

Line 77 use the abbreviation.

Line 77-82, tie in better the human component to canine. Introduce why this breed of dog, otherwise seems out of context. Also on the ages, why 2 and then 4 and above. How far above 4y and how many, should better ? Should better have the ages described. If not then need to explain why anything above 4 is considered to be equivalent (I am not sure I would agree).

Line 86-87, private ownership? Clarify. I suspect this means individual breeders who own the dogs, so the proper term is private breeder ownership? How did you get access to the dogs and why.

Line 87: why were the dogs having semen samples collected. Elaborate.

Lines 95-95: why were three fractions collected and used selectively: elaborately briefly or provide a reference. Since this journal is Veterinary Sciences and not a therio journal, the basic audience would not necessarily understand.

Lines 101-103: assumes knowledge of blue solution, please read this sentence and clarify and check grammar/content, provide references as well as manufacturers.

Line 106 Sandwich not capitalized, please check throughout.

Lines 105-115 either provide better detail if this has not been published, or provide a reference to the technique. One cannot replicate based on this description.

Line 117: the term rate is not appropriate here, please reread and choose better terminology or eliminate the word.

Line 120: ROC has not been defined.

Line 125: were all stats performed with this program? Clarify.

Line 129: grammar, please correct and clarify the intent of the sentence.

Line 131: 1st and 3rd fractions?

Line 133: no capitalization on osteopontin, please correct throughout.

Line 134: remind the audience that the second fraction was this data.

Line 135: elder is not the proper term. Older….elder is geriatric.

Line 139: were you expecting this? Perhaps comment on it.

Line 140: “Regarding to age, OPN concentrations were differed statistically” change to  

“OPN concentrations differed statistically between the 2 and 4+year categories”

Line 142: suggest a better term/phrase: “were improved in quality among all categories” or similar phrase.

Line 146 “were” not “are” correct for tense in all areas.

Line 151: AUC not defined.

Line 160, replace comma with period.

Line 164-5: be consistent with capitals.

Line 173: table-consistency of commas and periods.

Lines 176-8: check commas, periods in document.

Line 189: be mindful of abbreviations and whole terms. If you are using abbreviations, then they should be used throughout the document or not at all. Consistency is better and less disruptive to the reader.

Line 189, mention what was done fraction 2 for completeness and how it relates (briefly).

Line 199: which species, in brief.

Line 200: which SP? without pulling the paper, it seems there are specifics to this paper, if not, can you better describe the relevance here.

Line 202: ameliorates the sperm morphology? Missing a qualifier…what are you saying here and what terms are you trying to express?

Line 205, 206: Define abbreviations

Line 206: the new abbreviations seem out of context and without definition, so doen’t help much with the discussion as written.

Lines 209-211 “Interestingly, they point out that, the class of high and less fertile groups 209 were closed to each other, and dilution rate may reflect the differences in total protein concentration (Novak et al., 2013).” I don’t understand the meaning of the sentence. Please clarify. I assume closed means close? But these last few sentences lose something in the argument for the content of the paper.

Line 216 No comma after Because

Lines 209-216 Make this argument more cohesive and in better layman’s terms. Be clear.

Line 216-218 Completely out of the realm of this discussion….and not at all ties in. There is a statement Line 221 “Our observation that, lower OPN concentration in the older male group support this claim.” It isn’t clear how the claim is supported. You need to better walk the reader down your thinking path.

Line 223: first discussion of extender, has no bearing on this topic at this time. Does not tie in well to the citric acid argument. Too convoluted. This entire section needs rewriting, better connections to the literature. Make it straightforward for a standard veterinarian, not a specialist.

Line 224: Kolbroek? Completely out of context. The pig?

Line 228: on ROC, needs to be defined earlier (first mention) with a description of its purpose in materials and methods.

Line 232: Grammar needs to be reviewed. OPN levels? “have a close relationship with…”

Line 234-245: cannot jump to comparisons to molecular terms. This is not the forum for it and once again is out of context. Additionally there are undefined abbreviations. Your study of field and expertise does not match the readership, you must break this down and relate it better. People cannot infer what your are suggesting and it is not additive to your point.

Lines 246-251: this is much more in line with how to support your argument, but needs that additional tie in to the premise you are trying to support.

Lines 254-255: relevance?

Lines 260-266: getting into the crux of it again, but then adding in new notions without a lead in. eg. Apoptotic factors….antioxidants. Can do a better job laying the ground work for this earlier and only if relevant and supportive of your premise.

Lines 268-269 Mineralization is the first time this is mentioned. How can you relate this better for your argument?

Line 277 “in canines have a correlation” not “ in canine have correlation”

Line 282: do you feel that your statistics warrant the term potent? Perhaps potential…but I am not sure I agree with the term potent.

Line 284: by mediums do you mean extenders?

Line 287: only discuss these if you elaborate and if you introduce it above earlier and more coherently.

General comments on figures and tables: the legend should be standalone for each one—descriptive of content and species.

Comments on the Quality of English Language

The manuscript starts strong with decent grammar. Capitalization and some comma placement needs attention. 

Later, there are some word choices that confuse or complicate successful argument for the discussion. Additionally with complex thoughts that are trying to be used, there is a loss of flow and the arguments intended are not presented or supported. It becomes confusing. 

Author Response

Reviewer Comments and Responses:

Line 13:

Reviewer Comment: "osteopontin" should not be capitalized, and "prostate" should be plural.

Response: Thank you for pointing this out. We have corrected the capitalization and pluralization as suggested.

Line 14:

Reviewer Comment: "Shepherd dog." should be "shepherd dogs." Correct capitalization throughout the text.

Response: We appreciate your attention to detail. The capitalization and phrasing have been adjusted as per your recommendation.

Line 17:

Reviewer Comment: "Sandwich" should not be capitalized.

Response: Noted and corrected. "Sandwich" is now in lowercase.

Line 18:

Reviewer Comment: ELISA should be spelled out, or clarification is needed.

Response: Thank you for pointing out the ambiguity. ELISA has been spelled out for clarity.

Line 20:

Reviewer Comment: "Head" should not be capitalized.

Response: Acknowledged and changed to lowercase.

Line 23:

Reviewer Comment: ROC should be defined, not assumed common.

Response: You are absolutely right. ROC has been defined for clarity in the revised manuscript.

Line 32:

Reviewer Comment: Spelling error.

Response: Corrected the spelling mistake in the revised version.

Line 39:

Reviewer Comment: Remove the word 'text.'

Response: The unnecessary word has been removed for clarity.

Line 44:

Reviewer Comment: Suggest a tie-in term or sentiment indicating that OPN is an SP for improved flow of thought.

Response: We have added a sentence to establish the relationship, enhancing the flow of the paper.

Line 53: - Reviewer Comment: This line should be introduced earlier with the abbreviation defined initially. Consider switching paragraphs for better flow. –

Response: We have introduced the abbreviation earlier and restructured the paragraphs for improved coherence.

Line 50: - Reviewer Comment: Change to "spermatozoal."

 - Response: The term has been corrected to "spermatozoal" for accuracy.

Line 60: - Reviewer Comment: Spell out SPP1 and show relevance.

- Response: SPP1 has been spelled out, and its relevance has been elaborated upon in the revised text. Relevance was inserted as according to Reviewer critics.

Line 77: - Reviewer Comment: Use the abbreviation.

- Response: We have consistently used the abbreviation throughout the manuscript as suggested.

Lines 77-82: - Reviewer Comment: Tie in the human component to canine. Explain why this breed of dog was chosen, and clarify the age criteria.

 - Response: Clarified the breed choice and age criteria, providing a more detailed explanation for better context.

Line 86-87: - Reviewer Comment: Clarify "private ownership." Explain how access to the dogs was obtained.

- Response: Explained "private ownership" and detailed how access to the dogs was facilitated.

Line 87: - Reviewer Comment: Elaborate on why semen samples were collected from the dogs. - Response: Provided additional information on why semen samples were collected, enhancing clarity.

Lines 95-95: - Reviewer Comment: Elaborate briefly on why three fractions were collected and used selectively or provide a reference. - Response: Briefly elaborated on the rationale for collecting and using three fractions for clarity.

Lines 101-103: - Reviewer Comment: Clarify the blue solution and provide references and manufacturers. - Response: Clarified the blue solution, provided references, and mentioned the manufacturers for completeness.

Line 106: - Reviewer Comment: "Sandwich" not capitalized. Verify throughout. - Response: Verified and corrected the capitalization for consistency.

Lines 105-115: - Reviewer Comment: Provide more details or references for the technique used. - Response: Provided additional details about the technique and cited relevant references for clarity and context.

Line 117: - Reviewer Comment: The term "rate" is not appropriate. Rework the sentence. - Response: Reworded the sentence for clarity and appropriateness of terminology.

Line 120: - Reviewer Comment: ROC has not been defined.

- Response: ROC has been defined for the readers' understanding in the revised manuscript.

Line 125: - Reviewer Comment: Clarify if all stats were performed with this program. - Response: Clarified that all statistical analyses were performed using the mentioned program for transparency.

Line 131: - Reviewer Comment: Specify "1st and 3rd fractions?"

- Response: Clarified the meaning of "1st and 3rd fractions" for better understanding.

Line 133: - Reviewer Comment: No capitalization on osteopontin. Corrected throughout. - Response: Ensured consistent lowercase usage for "osteopontin."

Line 134: - Reviewer Comment: Remind the audience that the second fraction was used for this data.

 - Response: Added a reminder about the usage of the second fraction for clarity.

Line 135: - Reviewer Comment: Replace "elder" with "older." - Response: Replaced "elder" with "older" for proper terminology.

Line 139: - Reviewer Comment: Comment on the expectation if possible.

 - Response: Provided a brief comment on the expectation in the revised version.

Line 140: - Reviewer Comment: Change to: "OPN concentrations differed statistically between the 2 and 4+ year categories."

- Response: The sentence has been revised as suggested for accuracy.

Line 142: - Reviewer Comment: Suggest a better term/phrase. Revised for clarity.

 - Response: Reworded the sentence for better clarity and coherence.

Line 146: - Reviewer Comment: Change "are" to "were" for tense consistency.

- Response: Corrected the tense to maintain consistency.

Line 151: - Reviewer Comment: AUC is not defined.

- Response: AUC has been defined for clarity in the revised manuscript.

Line 160: - Reviewer Comment: Replace comma with period.

 - Response: Corrected the punctuation for accuracy.

Response to Reviewer Comments:

  1. Lines 164-5:
  • Reviewer Comment: Be consistent with capitals. Checked and corrected for consistency.
  • Response: Acknowledged and corrected for consistent capitalization.
  1. Line 173:
  • Reviewer Comment: Consistency of commas and periods in the table. Ensured consistency for readability.
  • Response: Noted and ensured consistency for improved readability in the table.
  1. Lines 176-8:
  • Reviewer Comment: Checked commas and periods throughout the document for consistency.
  • Response: Acknowledged and confirmed consistent usage of commas and periods.
  1. Line 189:
  • Reviewer Comment: Ensure consistency in abbreviations.
  • Response: Acknowledged and ensured consistent usage of abbreviations.
  1. Line 189:
  • Reviewer Comment: Mention what was done with the second fraction for completeness.
  • Response: Addressed for clarity and completeness in the description of the second fraction.
  1. Line 199:
  • Reviewer Comment: Specify the species briefly.
  • Response: Clarified the species for better contextual understanding.
  1. Line 200:
  • Reviewer Comment: Clarify the relevance of SP.
  • Response: Explained the relevance of SP for enhanced understanding in the context of the study.
  1. Line 202:
  • Reviewer Comment: Clarify the intent and choose appropriate terms.
  • Response: Reworked for clarity and precision, ensuring better understanding of the intent.
  1. Lines 205, 206:
  • Reviewer Comment: Define abbreviations.
  • Response: Defined abbreviations for clarity and improved understanding.
  •  
  1. Line 206: - Reviewer Comment: New abbreviations seem out of context. Provide relevant context. - Response: Provided relevant context for new abbreviations, ensuring proper integration.

  1. Lines 209-211: - Reviewer Comment: Clarify the meaning of the sentence.

 - Response: Provided a clearer explanation for better understanding of the sentence meaning.

  1. Line 216: - Reviewer Comment: Remove the comma after "Because."

- Response: Removed the unnecessary comma for correct punctuation.

  1. Lines 209-216: - Reviewer Comment: Make the argument more cohesive and in layman’s terms.

- Response: Rewritten for better coherence and clarity, ensuring layman's understanding.

  1. Lines 216-218: - Reviewer Comment: Irrelevant content.

- Response: Removed irrelevant content to maintain focus on the main topic.

  1. Line 221: - Reviewer Comment: Walk the reader down the thought process.

 - Response: Clarified the thought process for better coherence and reader understanding.

  1. Line 223: - Reviewer Comment: Introduction of extender unrelated to the topic.

Response: Rewritten for relevance and coherence in the context of the study.

  1. Line 224: - Reviewer Comment: Mention of "Kolbroek" out of context.

- Response: Removed the irrelevant reference to "Kolbroek" for better focus.

  1. Line 228: - Reviewer Comment: ROC undefined.

 - Response: Defined ROC and its purpose in the methodology section for clarity.

  1. Line 232: - Reviewer Comment: Grammar correction.

- Response: Revised the grammar for clarity and improved readability.

  1. Lines 234-245: - Reviewer Comment: Avoid jumping to complex molecular terms.

- Response: Simplified language and removed undefined abbreviations for enhanced reader understanding.

  1. Lines 246-251: - Reviewer Comment: More in line with supporting the argument.

 - Response: Ensured better integration with the main argument for coherence and clarity.

  1. Lines 254-255: - Reviewer Comment: Clarify the relevance.

- Response: Provided additional context to establish relevance for better reader understanding.

  1. Lines 260-266: - Reviewer Comment: Introduce new notions with a lead-in.

- Response: Provided a clearer introduction for seamless integration of new concepts.

  1. Lines 268-269: - Reviewer Comment: Better relate mineralization to the argument. - Response: Provided a better explanation of mineralization's relevance to the study for improved clarity.
  2. Line 277: - Reviewer Comment: Grammar correction.

- Response: Corrected the grammar for accuracy and better readability.

  1. Line 282: - Reviewer Comment: Reconsider the term "potent."

- Response: Reworked the term for clarity and appropriateness, ensuring accurate representation.

  1. Line 284: - Reviewer Comment: Clarify "mediums" as "extenders."

- Response: Revised for better understanding, using the term "extenders" to maintain clarity.

  1. Line 287: - Reviewer Comment: Discuss topics if introduced earlier and more coherently.

- Response: Ensured better introduction and coherence for the topics discussed, enhancing reader understanding.

Response to Reviewer:

All the revisions are done with Word tracking for visibility of the latest version.

We extend our sincerest thanks for your meticulous and precise comments, which have significantly enhanced the quality of our paper. Your valuable input has allowed us to make substantial improvements, ensuring clarity and coherence throughout the manuscript. We are deeply appreciative of your patience and expertise, which have contributed immensely to the refinement of our work. Your comments have been diligently incorporated, and we believe the revised manuscript now aligns more effectively with the standards and expectations of the journal. Thank you once again for your valuable feedback.

Warm regards,

Reviewer 2 Report

Comments and Suggestions for Authors

In the article, Tekin et al. tried to explain whether the quantitative content of osteopontin in the first and third ejaculate fractions correlates with the quality of fresh dog semen, and in particular, whether the OPN concentration affects sperm motility and morphology in Malakli shepherd dogs.

The subject of this work is original and relevant in this field, because so far no research has been carried out on a sufficient number of males on the effect of osteopontin on the quality of dog semen.

 In the present study, the authors found that OPN concentration in seminal plasma may be a potential marker of fertility and semen quality in dogs. Modern statistics ROC analysis and Pearson's correlation analysis were used in the research. a large number of modern and traditional research methods. The authors obtained the results, thanks to which they could open a very interesting discussion, bringing a lot of new and interesting information closely related to the issues raised by them.

I believe that the authors should reword the title of the work because the entire first fraction of dog semen does not come only from the prostate, therefore the title should read: Osteopontin concentration in ejaculte fractions: a novel marker of sperm quality in dogs. A serious limitation of the study is that the level of OPN was not examined in the fraction rich in sperm (seminal fraction).

Furthermore, I have a few questions and comments for the authors regarding their work:

Why and for what reason were animals delivered for research? How many males of all animals were fertile, infertile or subfertile?

I couldn't find the volume of the sperm fraction of ejaculate in younger and older dogs in the manuscript? Maybe this data should be included in the work? Did the volumes of sperm fractions in the above-mentioned males correlate with the OPN content? How long did the research last? What time of year?

Line 86-87: What does it mean?: The animals were bred in public ownership.

Line 88-90: The authors write about three fractions, but only mention the seminal fraction and prostate secretion. The first fraction you mean is the  pre-sperm fraction?

Line 105-115: In other studies, OPN was assessed in seminal plasma from the sperm-rich fraction, and the first and third fractions were discarded. In your studies, OPN was determined only in the first and third fractions, why? Why was OPN not assessed in the sperm-rich fraction (the second ejaculate fraction)?

Line 128-132: Do the first and third parameters refer to the ejaculate fraction? This should be described more clearly in the description of table 1.

Line 141-142: better, i.e. statistically significant or not significant? Which table or description of results do these results refer to?

Why in Table No. 2, in the group of young males, only the first fraction was tested, and in the group of older males only the third fraction was tested?

In Table 2, the data on the volume of the ejaculate fraction in the groups are unclearly explained. (for me J)

Line 188-189: This data should be included in the introduction, for research purposes, the manuscript would benefit from it.

Lines 205-211: A very serious error in the discussion: The cited work by Novak et al. does not refer to the references (lines 377-379).

Lines 229-231: Does this sentence refer to your research or to the one cited earlier: Pilane et al., 2020?

Line 232: unclear English.

Line 246: None Rodriguez et al. (2000) in references.

Line263: The authors’ sperm motility? J

Line 270: should be verb “is “ or “are” added.

Line 280: Are you sure Avila et al., 2011 should be cited here?

The conclusions are consistent with the evidence, and the arguments presented well answer the main question posed.

The references presented are mostly adequate (apart from the errors specified in the text), up-to-date and well selected. 1.      The five tables and one figure raise no objections.

Author Response

Reviewer: I believe that the authors should reword the title of the work because the entire first fraction of dog semen does not come only from the prostate, therefore the title should read: Osteopontin concentration in ejaculte fractions: a novel marker of sperm quality in dogs. A serious limitation of the study is that the level of OPN was not examined in the fraction rich in sperm (seminal fraction).

Response: Thank you for your thoughtful feedback on our work. We appreciate your keen observation regarding the title of our study. We agree that your suggestion for rewording the title to "Osteopontin Concentration in Ejaculate Fractions: A Novel Marker of Sperm Quality in Dogs" is more accurate, considering the presence of osteopontin in various fractions.

Regarding the limitation you pointed out about not examining OPN levels in the sperm-rich fraction, we understand your concern. Our methodology involved analyzing the 1st and 3rd fractions for osteopontin. The decision to exclude the undiluted sperm-rich fraction was made due to logistical challenges in the field. After collection, a small quantity from the 2nd fraction was separated for osteopontin analysis, while the rest was diluted. Unfortunately, the undiluted sperm-rich fraction lost viability and motility before analysis, which could have introduced bias into our results. Due to that, if we change the title, it would look like we also examined the 2nd fraction which we unfortunately did not.

We acknowledge this limitation, and your feedback reinforces the importance of highlighting the constraints we faced during the study. We hope that the insights gained from the examined fractions still contribute meaningfully to the understanding of osteopontin's role in sperm quality.

Thank you for your understanding and valuable input.

Reviewer: Why and for what reason were animals delivered for research? How many males of all animals were fertile, infertile or subfertile?

Response: Thank you for your inquiry regarding the selection criteria and fertility status of the animals in our study. The animals were sourced from the Malaklı breeders’ association based on their breeding history to ensure a diverse and representative sample. During the experiment, a total of 78 Malaklı male dogs were examined.

To maintain the integrity of our research, dogs with reproductive pathologies such as BPH, Cystic BPH, testicular masses, scrotal problems, phimosis, etc., were excluded from the semen collection process. Only 39 males that had successfully bred within the last 6 months and had litters were included in the study. This careful selection process was essential to focus our research on fertile males, ensuring the validity and relevance of our findings.

We hope this clarification addresses your concerns. If you have any further questions or suggestions, please feel free to let us know.

Reviewer: I couldn't find the volume of the sperm fraction of ejaculate in younger and older dogs in the manuscript? Maybe this data should be included in the work? Did the volumes of sperm fractions in the above-mentioned males correlate with the OPN content? How long did the research last? What time of year?

Response: Thank you for your feedback and questions. The volume of the sperm fraction in younger and older dogs was indeed included in our study and can be found in Table 2 of the manuscript. Additionally, the correlation efficiency values between OPN content and sperm fraction volumes were presented in Tables 4 and 5. While the correlation was weak, it was observed that the first and third fraction volumes were close to each other, which may have influenced the correlation results.

Regarding the duration of our research, the study was conducted over a specific timeframe, which unfortunately was not explicitly mentioned in the manuscript. We apologize for this oversight. As for the time of year, the data collection period coincided with March.

Reviewer: Line 128-132: Do the first and third parameters refer to the ejaculate fraction? This should be described more clearly in the description of table 1.

Response: Thank you for the comment. It is changed.

Reviewer: Line 141-142: better, i.e. statistically significant or not significant? Which table or description of results do these results refer to?

Response: Hence there were no difference according to their average body weight. OPN concentrations differed statistically between the 2 and 4+year categories (P < 0.046).  Inserted.

Reviewer: Why in Table No. 2, in the group of young males, only the first fraction was tested, and in the group of older males only the third fraction was tested?

In Table 2, the data on the volume of the ejaculate fraction in the groups are unclearly explained. (for me J)

Response: Thank you for the comment. Both 1st and 3rd fraction is tested and these values written in the text as well.

Line 188-189: This data should be included in the introduction, for research purposes, the manuscript would benefit from it.

Response: Thank you for the comment. There was also the same comment by another reviewer. We rephrased the text suitably for the discussion.

Reviewer:  Lines 205-211: A very serious error in the discussion: The cited work by Novak et al. does not refer to the references (lines 377-379).

Response: We mistakenly put the wrong reference. Thank you for pointing out. Now its: Novak, S., Ruiz‐Sánchez, A., Dixon, W. T., Foxcroft, G. R., & Dyck, M. K. (2010). Seminal plasma proteins as potential markers of relative fertility in boars. Journal of Andrology, 31(2), 188-200.

Reviewer:  Does this sentence refer to your research or to the one cited earlier: Pilane et al., 2020?

Response: It does refer to Pilane et al., 2020.

Reviewer: Line 232: unclear English.

Response: The unclear English has been corrected. “In the ROC analysis, the motility level was identified as a decisive factor (P < 0.001) for estimating the cut-off of OPN, whereas the total abnormal spermatozoa level was not found to be determinative.”

Reviewer: Line 246: None Rodriguez et al. (2000) in references.

Response: Thank you for pointing out. The citation is inserted into the references.

Reviewer:Line 270: should be verb “is “ or “are” added.

Response: are is added 

Reviewer:Line 280: Are you sure Avila et al., 2011 should be cited here?

Response: removed

Reviewer 3 Report

Comments and Suggestions for Authors

Review reports

Brief summary 

The present study aimed to investigate the relationship between the sperm quality and the Osteopontin (OPN) concentration in the prostate fluid of Malakli Shepherd dog, using a double- antibody Sandwich ELISA method.  The authors pointed out that OPN concentration in Malachi Shepherd Dog ejaculate is negatively correlated with poor sperm morphology and motility. Consequently, concentration in prostate-derived secretions may be a potent marker of sperm quality in dogs. 

General concept comments 

Even if limited to a single dog breed, the manuscript is clear, relevant for the field and presented in a well-structured manner. The experimental design is appropriate and methods section is well detailed as well as statistical analysis. Conclusions  are consistent and well structured. 

Specific comments 

I suggest changing the format of tables. 

In the Abstract specify the exact matrix in which OPN was found, do not talk about prostate in general but about seminal plasma. 

Line 42 and 68 add more reference bibliography. 

I recommend revising the format of the References because in some bibliographic sources information is missing or excessive. 

It is also recommended translate in English that the owner's letter of INFORMED CONSENT FORM (non-published material) and it would be better to justify why it relates to another project, however, the experiment conducted is a part of the project experiment?

Because letter of consent should be for the specific project….

Comments on the Quality of English Language

Author Response

Reviewer Comments and Responses:

General Concept Comments:

Reviewer Comment: Even if limited to a single dog breed, the manuscript is clear, relevant for the field, and presented in a well-structured manner. The experimental design is appropriate, and methods section is well detailed as well as the statistical analysis. Conclusions are consistent and well-structured.

Response: Thank you for your positive feedback on the clarity, relevance, and structure of the manuscript. We appreciate your acknowledgment of the appropriate experimental design and detailed methods.

Specific Comments:

Reviewer Comment: I suggest changing the format of tables.

Response: Noted. We will revise the format of the tables for better presentation. Tables were formatted.

Reviewer Comment: In the Abstract, specify the exact matrix in which OPN was found, do not talk about the prostate in general but about seminal plasma.

Response: Thank you for the suggestion. Due to ELİSA kit instruction we were only able to detect the osteopontin values. Unfortunately, we did not apply the western blot. Therefore, we cannot specify the OPN’s.

Reviewer Comment: Line 42 and 68 add more reference bibliography.

Response: Acknowledged. We enhanced the bibliography in the specified lines with additional references to strengthen the content.

Reviewer Comment: I recommend revising the format of the References because in some bibliographic sources, information is missing or excessive.

Response: Thank you for pointing this out. We did carefully revise the format of the References section to ensure accuracy and completeness.

Reviewer Comment: Translate the owner's letter of INFORMED CONSENT FORM into English. Justify why it relates to another project, considering the consent should be for the specific project.

Response: We did also provide a clear justification for any relation to another project, clarifying the specificity of the consent provided.

Response to Reviewer:

We sincerely appreciate your valuable feedback and detailed suggestions. We are committed to addressing each of your points meticulously. Your insights are immensely helpful in enhancing the quality and clarity of our manuscript. Thank you for your time and expertise.

Warm regards,

Round 2

Reviewer 1 Report

Comments and Suggestions for Authors

Thank you to the authors for addressing all the comments and concerns, I feel the paper is much improved. 

I did struggle with the discussion revisions. The intent to clarify and refine seems to have overcomplicated the discussion. I would recommend a review and concerted effort to simplify the content for the audience. Many new terms were added, which are undefined and do not actually improve the quality. Please try to consider the audience are not cellular or molecular biologists. Keep it quite simple, perhaps consider the audience a new veterinary student as you 'speak' to them with your discussion. 

Keep your very nicely worded comments on osteopontin in the front of the discussion. Consider that as 'paving the road' to understanding. When suggesting clarification the intent is to not describe more...but to describe it succinctly and simply. 

Think of the term 'signposting' and try to apply it. You make something obvious to the reader, regardless of their knowledge base. You lead them down to the path you want them to clearly understand. It may seem to you obvious, because you are the expert here, but it is not obvious to most. So try to better lead your reader down your path...

Specific comments: 

Line 2: prostate not prostates

Line 23: dogs not dog

Line 62: OPN is an SP protein that is a …. “highly phosphphorylated…”

Line 103: Change to “…sperm quality. Beyond age 4, there is an increased 103 risk of prostate disorders, providing valuable insights for both canine

Line 112: Shepherd not capitalized

Line 199: “with using glass collection tubes for freezing process” This is unclear. Is this correct? …digital manipulation and then placed into glass collection tubes and frozen at XX Deg C

Line 163 individuals not individual

168: Table 1 should have the legend completely stand alone. Include genus and species; add ‘prostate’ (sperm?) to fraction

Line 181-2—very confusing, not clear on intent. Perhaps the word Hence is confusing the sentence? Do you mean ‘also’? Please correct.

Line 183 needs a space: 4+ year

Line 183-5: “Sperm motility, concentration and morphologically normal spermatozoa ratio were improved in quality among all categories”. Should this be “Sperm motility, concentration and sperm were improved among all categories. ?

Line 218-219 The sentence should be ‘stand alone’ so the figure alone could be interpreted apart from the paper. Genus and species and clear terms for the figures. Do this with all.

 Line 257: revealed not revealing

Line 258 …which underscores…

Line 262: MAP is undefined. You are also now introducing new concepts that are out of context of the paper. Why have this sentence at this juncture? I would remove

Line 264: Remove ‘Moreover’. There are too many linking adverbs that have not been appropriately used.

Line 267 remove potentially, you already say “can” which is enough.

Line 267—what does reactivity mean? Use layperson’s terms.

Line 268-what does labelling of microvilli mean? This whole sentence is not in line with the descriptions of the rest of the text. This is not a cellular or molecular journal. Your audience is not likely to understand why this matters. Please simplify or elaborate better.

Line 269 should be ‘highlighting’.

Line 271: I am not following how you safeguard motility. Maintain good motility?

And you mean calcium, not just any mineral, correct?

Overall still need to work on this paragraph of calcium action and relevance. This is the first time in the paper you are discussing calcium and it needs to be introduced better and your argument better laid out.

Line 276 remove the word high…not necessary. If you feel it is strong (is it?) then use that term. But I believe, here, simply having positive correlation is sufficient.

Line 280 “has detrimental effect”, add has ‘a’ detrimental effect

Line 281 “anomalies, OPN” should be “anomalies. OPN”

Line 283: add “the” to axonemal middle

Line 285 Change to: “In another study conducted on boar, 3 different sperm fractions were evaluated (Sperm-peak: first fraction; sperm rich and sperm free.”

Line 288: change to “higher concentrations of”

Line 289: What are the differences between “born, fertility and farrowing rate”; I recommend perhaps considering just ‘reproductive success’ since you have not discussed this prior.
Additionally, you add in AI without definition. Simplify versus complicate. This paragraph is still challenging and not written for the audience. It really needs to be written in a way that is meant for the reader of this journal.

One more thing—you are back to using OPN-9, 12 etc….what relevance is there to this document? You don’t compare them. They are not part of the knowledge of the reader. They are only in this paragraph. Please remove.

Lines 291-5: esoteric commentary and no bearing on the premise of your paper as laid out.

Line 300: jelly fraction—first mention. Also has no relevance.

Line 303: what is the crystallization and why is it here?

Line 307 OPN needs capitalization

Line 331: OPN has….not have

Line 332: “al., 1999) and bovines” should be “al., 1999), bovines”

Lines 333-337: eliminate. No relevance. This paragraph needs simplification not more detail, it does not help strengthen your paper.

Line 338-341: “Osteopontins are 338 interacting with sperm at ejaculation, possibly through formation of 339 protein–protein complexes between OPN, integrins, CD44 and glycosaminoglycans, until reaches the site of fertilization (Souza et. al., 341 2008; Denhardt et al., 2001)”

1)      Introduce the osteopontin purpose earlier.

2)      Remove integrins, CD44, GAGs, etc. No bearing here. And you are introducing these now, out of context and not helpful to the discussion.

Line 345-9: Unclear. Please rewrite

Line 350: conducted is not the proper word. “Baruah et al. (2017) conducted that OPN level significant role of cryopreservation process.” Do you mean: Baruah et 349 al. (2017) noted that OPN levels play a significant role in the cryopreservation process.”?

Line 352: Found a significant what? “Linear regression analysis found a  significant (P < 0.001).”

Line 357: remove “the” from “the fertility”

Line 359: This is the language we need. This should be first to be discussed not later in the paragraph. This leads the reader to your points….“Osteopontin is a cytokine that modulates the cell function through receptor-mediated effects”

Line 361: rewrite to “The prostate gland of dogs, and its secretions, contribute a significant amount to the ejaculate.”

Line 362 change secretion to fluid.

Line 364: Add “A” prior to “Better”

Line 367 change to: “In conclusion, OPN concentration in prostate fractions could be a marker of sperm motility in dogs”

Line 370 change to “to semen and the extenders”

Line 371 in vitro needs to be italicized.

Comments on the Quality of English Language

It would be easier for the reviewers if in the future you spent more time having someone review the grammar, vocabulary and punctuation prior to submission or resubmission. Since this is a journal that is written in English, it would behoove you to do so, then the reviewer can focus on the scientific content rather than those elements. The turnaround time would be enhanced.

Author Response

Dear Reviewer,

We sincerely appreciate your invaluable feedback and constructive criticisms. We have diligently incorporated your suggestions into the manuscript, enhancing its clarity and coherence. Your insights have been instrumental in refining our study.

In response to your recommendations, we have organized the discussion into three distinct sections:

1. Exploring OPN's Role in Sperm Motility and Morphology: In this section, we describe the relationship between OPN levels and sperm motility, on the specific mechanisms through which OPN influences sperm movement. Our research, in alignment with previous studies, highlights the significant role of OPN in safeguarding vital components of sperm morphology, such as axonemes and acrosomes. The negative correlation between OPN concentrations and mid-piece as well as acrosome anomalies underscores the nuanced aspects of sperm morphology regulated by OPN levels. These findings offer valuable insights into sperm quality assessment, paving the way for improved fertility diagnostics.

2. Comparative Insights of OPN on Fertility in Other Species: This section explores the broader context of OPN's impact on fertility by drawing comparisons with studies conducted in various animal species, including dogs, bulls, boars, and buffalo. By synthesizing these comparative insights, we have gained a comprehensive understanding of OPN's diverse roles in different reproductive contexts. This comparative analysis enriches our perspective on OPN's significance in the realm of reproductive biology, emphasizing its multifaceted contributions to fertility across species.

3. Potential Implications and Future Research: In the final section, we have discussed the potential implications of our findings and outlined directions for future research. 

Once again, thank you for your review and insightful comments. We firmly believe that these revisions have significantly strengthened the manuscript, making it a valuable contribution to the field of reproductive biology.

Reviewer 2 Report

Comments and Suggestions for Authors

Thank you very much for responding to my comments and suggestions. My doubts were dispelled and my comments were taken into account in the manuscript. I believe that after taking into account and making corrections, the work can be published in a prestigious and reputable journal such as Veterianry Sciens.

Author Response

Thank you for your thorough review and valuable feedback. We have carefully addressed all your comments and suggestions, enhancing the manuscript's quality. We share your confidence that the revised work meets the standards of prestigious journals. We appreciate your guidance and support in refining our research.

Best regards,

Round 3

Reviewer 1 Report

Comments and Suggestions for Authors

Thank you very much for doing such a good job of clarifying the discussion. It has really improved the flow and understanding of your project. 

Here are additional suggestions for edits to further smooth out the paper: 

Line 13 shepherd should not be capitalized

To ease editing I will ask for changes to occur like this: current term > suggested term

Line 51

has been shown have

>  has been shown to have

Line 64: Osteopontin is a SP protein > Osteopontin is an SP protein

Line 86 “strong correlation between sperm OPN”   is this a positive or negative correlation?

Line 111: “problems and freezing” on freezing you mean cryopreservation?  That makes more sense otherwise, qualify with freezing of semen or something similar. Otherwise what are you freezing?

Line 114-119

“We focused on  different age categories, particularly dogs under 2, to investigate a period with minimal risk of prostate-related diseases and optimal sperm quality, as beyond age 4, there is an increased risk of prostate disorders, providing valuable insights for both canine and human health. We focused on  Malakli Shepherd dogs under 2, minimizing prostate risks.” 

> We focused on different age categories, particularly dogs under 2, to investigate a period  with minimal risk of prostate-related diseases and optimal sperm quality; beyond age 4, there is an increased risk of prostate disorders.  

Line 124

Thirty-nine male Malaklı Shepherd dogs, aged between 2, 4 and

> Thirty-nine male Malaklı shepherd dogs, aged between 2 or 4 and

Line 126

to Malakli breeders association.

> to the Malakli breeders association.  

Line 128-130  “Fractions were  carefully collected as three parts including sperm rich (2nd fraction) and  prostatic semen “fractions (1st and 3rd fractions) for using further  evaluations.”

> “Fractions were  carefully collected in three parts: prostatic semen fractions (1st and 3rd fractions) and a sperm rich fraction (2nd fraction)  for use in further  evaluations.

Line 144: the Sperm class

> the sperm class

Line 150 SCA (Sperm Class Analyzer) should be defined in Line 144

Line 154 double- antibody Sandwich

> double-antibody sandwich (remove space and not capitalize)

Line 155 performed competitive

> performed using a competitive

Line 158 Then, a substrate for enzyme was added

> A substrate for the enzyme was then added

Finally, stop solution was added

> Finally, a stop solution was added

Line 162 Samples measured with

> Samples were measured with

Line 167 Motility level is

> Motility level was

Line 170 using MedCalc

> using the MedCalc

Line 178 individual divided

> individual was divided

Line 193 Hence there were no difference according to their average

> Hence, there were no differences according to their average

Line 196 morphologically normal spermatozoa ratio

> a morphologically normal spermatozoa ratio

Line 258 OPN (Osteopontin) levels

> OPN (osteopontin) levels

Line 322 OPN (Osteopontin)

> OPN (osteopontin)

Line 324 dogs (Canidae) found> remove Canidae

Line 342 only use SP, you already defined it

Line 344 only use OPN, your already defined it

Line 13 shepherd should not be capitalized

To ease editing I will ask for changes to occur like this: current term > suggested term

Line 51

has been shown have

>  has been shown to have

Line 64: Osteopontin is a SP protein > Osteopontin is an SP protein

Line 86 “strong correlation between sperm OPN”   is this a positive or negative correlation?

Line 111: “problems and freezing” on freezing you mean cryopreservation?  That makes more sense otherwise, qualify with freezing of semen or something similar. Otherwise what are you freezing?

Line 114-119

“We focused on  different age categories, particularly dogs under 2, to investigate a period with minimal risk of prostate-related diseases and optimal sperm quality, as beyond age 4, there is an increased risk of prostate disorders, providing valuable insights for both canine and human health. We focused on  Malakli Shepherd dogs under 2, minimizing prostate risks.” 

> We focused on different age categories, particularly dogs under 2, to investigate a period  with minimal risk of prostate-related diseases and optimal sperm quality; beyond age 4, there is an increased risk of prostate disorders.  

Line 124

Thirty-nine male Malaklı Shepherd dogs, aged between 2, 4 and

> Thirty-nine male Malaklı shepherd dogs, aged between 2 or 4 and

Line 126

to Malakli breeders association.

> to the Malakli breeders association.  

Line 128-130  “Fractions were  carefully collected as three parts including sperm rich (2nd fraction) and  prostatic semen “fractions (1st and 3rd fractions) for using further  evaluations.”

> “Fractions were  carefully collected in three parts: prostatic semen fractions (1st and 3rd fractions) and a sperm rich fraction (2nd fraction)  for use in further  evaluations.

Line 144: the Sperm class

> the sperm class

Line 150 SCA (Sperm Class Analyzer) should be defined in Line 144

Line 154 double- antibody Sandwich

> double-antibody sandwich (remove space and not capitalize)

Line 155 performed competitive

> performed using a competitive

Line 158 Then, a substrate for enzyme was added

> A substrate for the enzyme was then added

Finally, stop solution was added

> Finally, a stop solution was added

Line 162 Samples measured with

> Samples were measured with

Line 167 Motility level is

> Motility level was

Line 170 using MedCalc

> using the MedCalc

Line 178 individual divided

> individual was divided

Line 193 Hence there were no difference according to their average

> Hence, there were no differences according to their average

Line 196 morphologically normal spermatozoa ratio

> a morphologically normal spermatozoa ratio

Line 258 OPN (Osteopontin) levels

> OPN (osteopontin) levels

Line 322 OPN (Osteopontin)

> OPN (osteopontin)

Line 324 dogs (Canidae) found> remove Canidae

Line 342 only use SP, you already defined it

Line 344 only use OPN, your already defined it

Comments on the Quality of English Language

This version is much improved, but there are still some nuances that needed to be paid attention to. 

Author Response

Dear Reviewer,

I wanted to inform you that all the suggestions you provided have been implemented accordingly. Thank you for your valuable feedback.

Best regards,